# Performance of Short Fiber Interlayered Reinforcement Thermoplastic Resin in Additive Manufacturing

**DOI:** 10.3390/ma13122868

**Published:** 2020-06-26

**Authors:** Congze Fan, Zhongde Shan, Guisheng Zou, Li Zhan, Dongdong Yan

**Affiliations:** 1Department of Mechanical Engineering, Tsinghua University, Beijing 100084, China; fcz15@tsinghua.edu.cn (C.F.); zougsh@tsinghua.edu.cn (G.Z.); 2State Key Laboratory of Advanced Forming Technology and Equipment, China Academy of Machinery Science & Technology, Beijing 100044, China; 13810032191@139.com (L.Z.); ncutyandd@163.com (D.Y.)

**Keywords:** 3D printing, thermoplastic resin, additive manufacturing, composite

## Abstract

To further improve the mechanical properties of thermoplastic resin in additive manufacturing (AM), this paper presents a novel method to directly and quantitatively place the short fibers (SFs) between two printing process of resin layers. The printed composite parts with SFs between the layers was reinforced. The effects of single-layer fiber content, multi-layer fiber content and the length of fibers on the mechanical properties of printed specimens were studied. The distribution of fibers and quality of interlayer bonding were assessed using mechanical property testing and microstructure examination. The results showed that the tensile strength of the single-layered specimen with 0.5 wt% interlayered SFs increased by 18.82%. However, when the content of SFs continued to increase, the mechanical properties declined because of the increasing interlayered gap and the poor bonding quality. In addition, when the interlayered SFs length was 0.5–1 mm, the best reinforcement was obtained. To improve the interfacial bonding quality between the fiber and the resin, post-treatment and laser-assisted preheating printing was used. This method is effective for the enhancement of the interfacial bonding to obtain better mechanical properties. The research proves that adding SFs by placement can reduce the wear and breakage of the fibers compared to the conventional forming process. Therefore, the precise control of the length and content of SFs was realized in the specimen. In summary, SFs placement combined with post-treatment and laser-assisted preheating is a new method in AM to improve the performance of thermoplastic resin.

## 1. Introduction

Thermoplastic resin has the advantages of high humidity resistance, high corrosion resistance and good forming performance. [1,2,3]. It has been promoted as a recyclable environmentally friendly materials. In recent years, thermoplastic resins have been widely used in the field of fused deposition modeling [2]. Many researchers have attempted to use various methods to improve the mechanical properties of printed parts.

Initially, short fibers (SFs) were added to thermoplastic resin filaments, causing blocking and transfer effects, and a mutagenic effect to prevent crack propagation through the interface. Thus, the fibers bear the pressure of the specimen, and this changes the optical, acoustic, thermal and impact resistance of the forming parts [4,5,6]. Experiments have found that the fiber orientation in the fiber-forming direction is as high as 91.5% [4]. The fiber characteristics such as the length, content and printing parameters affect the mechanical properties of the forming specimen [7]. By optimizing the printing parameters, the interfacial bonding quality, the impregnation degree of the fiber and resin, and the crystallinity of the resin matrix can be improved. When SFs are added to the resin filaments, the mechanical properties of printed parts can be improved by optimizing the forming parameters and controlling the orientation and distribution of the fibers [8]. While the short fibers are added in filaments, the fiber orientation, fiber distribution and fiber length are affected by the forming process of filaments. In general, the filaments are made by a screw extrusion machine. Karsli et al. [9] added 13 mm carbon fibers to the resin, and observed through image processing that the fiber length in the printed part was concentrated at 0–50 μm. This indicated that a large number of fibers fractured during the filament forming process. As the fiber content increases, the degree of fiber breakage increases, and the length of the fiber in the specimen decreases. The reason for this is that the high shear mixing in the screw extrusion filament forming process leads to the wear and breakage of fibers. With increasing fiber content, the friction and wear between fibers increase, which further accelerates fiber breakage. The length and distribution of fibers are also affected, which restricts the improvement of the mechanical properties.

After this work, some scholars tried to modify the resin matrix with reinforcements, such as graphene and carbon nanotubes. The addition of one-dimensional and two-dimensional carbon structures enhances the matrix strength and improves the mechanical properties of the composite [10,11].

In recent years, continuous fibers have been added to the thermoplastic matrix to work as pre-preg tape, which imposes a pressure on the fiber interface through matrix. Therefore, the tensile and flexural properties of the printing specimen are improved 5-10-fold [12,13,14]. Previously, researchers have focused on the printing process and forming mechanism [15,16,17,18]; although this has a good reinforcement effect in AM, the limitations of the poor interfacial and interlayer bonding in the continuous fiber composite still require further research and improvement [19]. Thus, a low-cost, easy-to-use method for enhancing the mechanical properties in AM is necessary.

In this paper, SFs are added between the printing layers in a quantitative way to improve the poor mechanical properties of parts printed with pure thermoplastic resin. The effects of parameters, such as the fiber length, content and number of layers deposited on the mechanical properties of the specimens are studied. The distribution and bonding state of the fibers between the layers are revealed to analyze the reinforcement mechanism of the SFs between the layers. To further improve the bonding quality between the fibers and the resin, post-treatment and laser-assisted preheating are carried out during the additive manufacturing. Then, the improvement of the interfacial bonding quality and mechanical properties of the composite are studied, and the effects of different laser powers on the mechanical properties are investigated. The interlayer placement technology of SFs is used in the process of additive manufacturing, which can realize the fiber addition and enhance the performance at specific positions in the composite structure.

## 2. Materials and Methods

### 2.1. Materials for Testing and Research

There are many types of thermoplastic resin. In this study, polylactic acid (PLA) with good biodegradability was selected as the resin matrix [20]. PLA is a common engineering plastic with a melting point of about 175 °C, a softening point of about 60 °C, a thermal degradation temperature of 250 °C and a difference of 75 °C between the melting flow temperatures, thus providing a wide selection of printing temperatures [21]. The short carbon fibers were cut from continuous carbon fibers produced by TORAY Corporation of Japan, and the sizes of short carbon fibers were 0.1, 0.5, 1, and 2 mm.

### 2.2. 3D Printing Equipment with Fiber Placement

In this study, the AM-FW300 (forming size 300 mm × 300 mm × 300 mm) 3D printing machine, which was developed by China Academy of Machinery Science and Technology, was used to form the test specimen, as shown in Figure 1. Firstly, the 3D printer printed the layer *n* in the print bed. Secondly, a fixed amount of short fibers was placed on the layer *n* by the fiber sprayer. Finally, the layer *n*+1 was printed to cover the interlayered short fibers. The short fibers are only placed when required by the strategy. Figure 2 is the schematic diagram of the interlayer placement of SFs. The fiber sprayer is the key to achieve a quantitatively and homogeneous placement so that a sprayer equipment can be developed, as shown in Figure 1. During the forming process, parameters, such as the diameter of the nozzle, temperature of liquefier, substrate temperature, printing speed, hatch spacing, layer thickness and print path all affect the accuracy and performance of the final parts. After the preliminary optimizing experiment for processing parameters, the printing parameters shown in Table 1 were selected for this study.

### 2.3. Performance Test and Cross—Section Observation

To characterize the performance of the specimen accurately, tensile specimens were printed by the GB/T 1447–2005–standard. The dimensions of the specimens are shown in Table 2. The GB/T 1449–2005 standard was used to print three-point flexural test specimens. To ensure the stability of the test results, five specimens of each group above were printed and measured under the same testing conditions [22,23]. Figure 3 shows the schematic diagram and physical picture of the mechanical testing specimen. Figure 3a shows the internal structure of the printed specimen with a single layer of SFs. The specimen as divided into three different areas: the shell, the PLA layer, and the short carbon fiber reinforced printing layer (SCFRP).

The US–INSTRON–5567 universal testing machine (Instron, Norwood, MA, USA) was used in the tensile and flexural testing. The cross section of the specimen was analyzed by a JEOL JSM–7500F (JEOL, Tokyo, Japan, 10 kV) scanning electron microscope to observe the void and fiber/resin bonding effect in the fracture section of the test specimen. The grain size of the post-treatment specimen was measured by a Rigaku 12 kW X–ray diffractometer (D/MAX–RB, Rigaku, Tokyo, Japan) with a scan range of 5–60° and speed of 2°/min.

### 2.4. Post-Treatment

PLA resin is a semi-crystalline material. Marcus et al. [24] found that when the annealing temperature is 80 °C, the crystallinity of the specimen reaches a maximum value within 30 min, and the strength and hardness of the specimen increase while the crystallinity increases. Therefore, a post-treatment time of 2 h was selected in this study to ensure that the crystallinity of the specimen was obtained. The annealing treatment was arranged after the printing. The specific parameters are shown in Table 3. Four different temperatures were chosen based on the glass transition temperature. There was air at a pressure of 1 atm in the oven, and the printed specimens had no dimensional constraints in the oven.

### 2.5. Laser—Assisted Preheating

A 915 nm fiber laser with a maximum power of 30 W was used as the laser source for pre-heating. Figure 4 shows a schematic diagram of the additive manufacturing with integrated laser preheating function. The laser was installed on the side of the traditional heating nozzle, and the laser beam could be projected onto the forming platform to preheat the SFs. The carbon fiber was black, which facilitated heating. The laser power was changed to meet the required laser intensity, which further improved the heating efficiency of the laser aimed at the resin surface. The wavelength of the guide beam was 658 nm, the focal length of the laser was 45 mm, and the diameter of the projected spot at this distance was 1 mm. The laser was turned off during the printing process of pure resin and it was turned on after the placement step; then, it was driven by the three-axis motion system to print so as to achieve interlayer heating according to the scanning path.

### 2.6. Short Fibers Reinforcement Mechanism

Figure 5 shows the stress state of the SFs in the resin matrix. When the load acts on both ends of the composite material, the stress is transmitted to the SFs through the interface between the fiber and the matrix. This is determined with the balance of the force:(1)$$\pi r^{2}\sigma_{f}+2\pi r\cdot dz\cdot \tau =\pi r^{2}\left(\sigma_{f}+d\sigma_{f}\right)$$
where *σ_f_* is the axial tensile stress of the fiber, *τ* is the shear stress on the fiber surface, r is the radius of the fiber, and the fiber is parallel to the *z* direction. As the high stress concentrates at the end of the fiber, it can be considered that the two sides of the matrix have yielded and separated and the normal stress is 0. Assuming that the matrix around the fiber is completely plastic, the shear stress does not change with the shear strain. It can be considered that the shear stress is a constant; thus:(2)$$\sigma_{f}=\frac{2\tau_{\gamma }z}{r}$$

At the middle of the fiber, where *z* = *L*/2, the tensile stress of the fiber reaches the extreme value (*σ_f_*)*_max_*:(3)$${\left(\sigma_{f}\right)}_{max}=\frac{\tau_{y}L}{r}$$
where (*σ_f_*)*_max_* is related to the external load, *σ_c_* is acting on the composite, assuming that the fiber and the matrix are tightly coupled and deform simultaneously, i.e., the strain is equal, then:(4)$${\left(\sigma_{f}\right)}_{max}=\sigma_{c}\frac{E_{f}}{E_{c}}$$
where *E_f_* and *E_c_* are the elastic modulus of fiber and resin, respectively, and the above formula is
(5)$$\frac{L_{f}}{r}=\frac{\sigma_{c}E_{f}}{E_{c}\tau_{\gamma }}$$
(6)$$\frac{L_{c}}{r}=\frac{\sigma_{fu}}{\tau_{\gamma }}$$
where *L_f_* is the transmission length of the load, *L_c_* is the critical fiber length, *σ_fu_* is the ultimate strength of the fiber, and the physical meaning of *L_c_* is the corresponding minimum length of the fiber when it exerts the maximum load-bearing efficiency. Therefore, the average stress of the fiber in the unidirectional short fibers composite material is:(7)$$\bar{\sigma_{f}}=\frac{{\left(\sigma_{f}\right)}_{max}}{2}\;\;(L_{f}<L_{c})$$
(8)$$\bar{\sigma_{f}}={\left(\sigma_{f}\right)}_{max}\left(1-\frac{L_{c}}{2L_{f}}\right)\;\;(L_{f}>L_{c})$$

When the transfer length *L_f_* is less than the critical fiber length *L_c_*, the fiber’s maximum stress is less than its average breaking strength, and the failure of the composite material depends on the failure of the matrix or the interface. When the transfer length *L_f_* of the fiber load is greater than the critical fiber length *L_c_*, the fiber can be maximized. The breaking of the composite material depends on the breaking of the fiber.

Ignoring the Poisson effect during the stress of the composite material, the strength of the composite material can be predicted by the mixing law of weighted average:(9)$$\sigma_{c}={\bar{\sigma }}_{f}V_{f}+\sigma_{m}V_{m}$$
(10)$$\sigma_{c}=\frac{{\left(\sigma_{f}\right)}_{max}}{2}V_{f}+\sigma_{m}V_{m}\;\;(L_{f}<L_{c})$$
(11)$$\sigma_{c}={\left(\sigma_{f}\right)}_{max}\left(1-\frac{L_{c}}{2L_{f}}\right)V_{f}+\sigma_{m}V_{m}\;\;(L_{f}<L_{c})$$
where *V_f_* and *V_m_* are the volume fraction of the fiber and the matrix, respectively. ${\bar{\sigma }}_{f}$ and *σ_m_* are the average stress of the fiber and the matrix stress when the composite fails.

## 3. Results and Discussion

### 3.1. Effect of Short Fiber Content on Performance of the Additive Manufactured Thermoplastic Resin Parts

#### 3.1.1. Mechanical Properties Test

The concentric circle path filling method was chosen to additively manufacture three layers of pure PLA resin. The SFs were laid on the fourth layer; then, the printer continued to print three layers of pure PLA resin to obtain composite test specimen with a single-layer of SFs. Figure 6 shows the stress-strain curves of specimens with different fiber contents. The fiber content range is 0–2 wt% and the fiber length is 0.5 mm. Since five specimens of each group are printed, the measurement curve is close to the average tensile property value, which is selected and shown in Figure 6. Figure 7 shows the testing results of mechanical properties of specimens with different fiber contents. The testing results of each mechanical property are represented by a box chart, and the square covers 25–75% confidence intervals. The red line in Figure 7 is the average of five measurements under the same conditions, which is used to indicate the trend of the effect of fiber content on mechanical properties. In the following sections, the stress-strain curve is drawn in the same way, and the box diagram is defined in the same way.

In Figure 6, the tensile specimens of composites with different fiber contents undergo linear elastic deformation until breakage, and there is almost no yield phenomenon. The fiber contents of the single layer in Figure 7a are 0 wt%, 0.5 wt%, 1 wt%, 1.5 wt% and 2 wt%, respectively. When the fiber content increases from 0 wt% to 2 wt%, the tensile strength increases before decreasing. When the fiber content is 0.5 wt%, the tensile strength reaches the extreme value of 37.76 MPa, which increases by 18.82% compared to pure resin, at 30.65 MPa. A *t*-test was used to compare the tensile strength of the two groups (*p* = 0.003, *p* < 0.05 and *p* < 0.01, respectively), which indicated that a statistically significant difference exists between the results; in particular, when the fiber content accounts for more than 1.5 wt%, the tensile strength of the specimen decreases sharply. When the fiber content is 2 wt%, the average tensile strength is 24.65 MPa, which is 19.61% lower than that of pure resin material. The effect of carbon fiber content on Young’s modulus is shown in Figure 7b. As the fiber content increases from 0 wt% to 0.5 wt%, the Young’s modulus of the specimen increases to an extreme value of 1.57 GPa, and then decreases sharply with increasing fiber content. The changing rule of the Young’s modulus of the specimen is not obvious. It slightly changes around 4%, which is between 1 wt% and 2 wt% of fiber content. The three-point flexural test was used for testing the effect of fiber content on flexural performance. The test results are shown in Figure 7c. Upon the addition of fibers, the flexural strength increased. When the SFs content laid between the layers is 1 wt%, the flexural strength reached the extreme value of 98.81 MPa, which was 22.91% higher than 80.38 MPa, the flexural strength of the pure resin specimen. A *t*-test was used to compare the flexural strength of the two groups (*p* = 0.012, *p* < 0.05). It indicates that a statistical difference exists between the results. However, the range of the flexural strength of the specimen with the fiber content between 0.5 wt% and 1.5 wt% changed very slightly, at only 5%. When the fiber content was more than 1.5 wt%, the flexural strength decreased rapidly. The flexural strength of the specimen with a fiber content of 2 wt% was 68.55 MPa, which is lower than that of pure resin. Figure 7d shows the influence of the content of the single-layer fiber on the elongation at break of the specimen. The fiber content is 0–1.5 wt%, and the change of the elongation at break is not obvious. When the fiber content is 2 wt%, the elongation at break decreases rapidly.

According to the mixing law of composite materials, the mechanical properties of shaped parts increase linearly with the increase of fiber content. In this study, to further explore the effect in SFs content on the mechanical properties of the specimen, the total content of SFs in the specimen was changed by altering the number of the deposited SFs layers. The content of the single-layer fiber is 0.5 wt%, and the length is 0.5 mm. In this experiment, 1–5 layers of SFs were deposited. The specific placement strategy is shown in Figure 8. During the printing process, there were two different interfaces in the test piece, namely the fiber-resin interface and the resin-resin interface. In Figure 8, the green layer is the resin layer and the gray layer is the SFs layer. The tensile strength testing results are shown in Figure 9. Figure 9a,b show that when the number of layers increased from one layer to two layers, the tensile strength of the specimen increases from 37.76 MPa to 39.37 MPa, and the Young’s modulus increased from 1.57 GPa to 1.62 GPa. A *t*-test was used to compare the tensile strength of the two groups (*p* = 0.022, *p* < 0.05). It indicated that a statistical difference exists between the results. The increase in tensile properties and the increase in SFs content is disproportionate. In addition, when the number of SFs deposits is more than two, the tensile properties decrease rapidly, and when five fibers were deposited, the tensile strength decreased to 33.64 MPa.

#### 3.1.2. SEM Observation

The fracture interface of the tensile test specimen was observed by SEM, and the effect of the content on the porosity and the interfacial bond between the fiber and the thermoplastic matrix was analyzed. As shown in Figure 10a,c,e, there are special triangle voids of 3D printing inside the specimen. The voids are determined by the forming process parameters, especially the width of the printing raster. SFs have good dispersion inside the specimen, and the orientation in the horizontal direction among the layers is random. As shown in Figure 10a,b, when the fiber content is 0.5 wt%, the interlayer bonding is good. Figure 10b shows that the SFs are dispersed in the matrix, a large number of fibers broke during the specimen stretching process, and a small amount of fibers was pulled out. When the interlayer fiber content continued to increase to 1.5 wt%, as shown in Figure 10c,d, delamination began to occur between layers, and some fibers were not effectively impregnated with the matrix but dispersed at the fracture interface. As shown in Figure 10e,f, when the fiber content is 2 wt%, the delamination is obvious, and a large amount of delamination occurs. Cracks appeared between the layers, and a large number of fibers pulled off from the layers, dispersed at the fracture interface, and did not play the role of bearing stress.

#### 3.1.3. Discussion

Adding SFs to the thermoplastic resin filament can improve the mechanical properties of the printed specimens. In this paper, the SFs are directly laid and added between the printing layers. The mechanical testing results show that the strength of the specimen is improved. As shown in Figure 7, when the fiber content between layers is 0.5 wt%, the tensile and flexural properties of the specimen are better than pure resin materials. The load is transferred to the high strength and modulus fibers through the resin matrix to improve the strength of the composites. The cross section of the specimen after tensile fracture is shown in Figure 10a. The fiber breakage at the interface indicates that the main failure mode of the material is fiber breakage. The fiber breaks instantly when it is subjected to ultimate stress, causing the stress of the adjacent resin material to increase suddenly, as the stress is much higher than the resin can withstand. Due to the limited local plastic strain, a macro-brittle failure behavior occurs, as shown in Figure 6. The results in Figure 7 indicate that the mechanical properties of the composite material did not increase with the increasing of the short fiber content. When the fiber content increases to 1.5 wt% and 2 wt%, the delamination phenomenon in the fracture section of the specimen is serious, as shown in Figure 10c,e, and there are more interlayer gaps. As the fiber is added as a “cold state”, it only acts as a reinforcement when it is combined with the “high energy state” molten resin in the printing process. Due to the presence of the thermal pressing device, certain amounts of SFs can bond with the matrix. However, an excessively high fiber content causes the molten resin to rapidly cool and condense on the surface and fail to produce a coating effect. This will lead to voids and “free” carbon fibers being formed, causing splitting damage to the substrate and deteriorating the mechanical properties.

In this paper, the number of SFs layers is changed to increase the total content of SFs in the test specimen. However, when the number of layers is increased from one layer to two layers, as shown in Figure 9a,b, the improvement in tensile properties is not obvious. The increase is not proportional to the increase in SFs content. When the number of layers further increases, the tensile properties decreases. The reason for this is that the SFs layer is additive, which affects the surface flatness and even the number of internal voids, leading to the poor bonding of the two upper and lower resin layers adjacent to it. When laying a single layer of fibers, the printing of multiple resin layers can make up for the defects caused by the deposited fiber layers. However, the stacking of multiple fiber placement layers will aggravate the accumulation of defects, affecting the melt flow of resin and the coating impregnation of fibers. As the SFs content increases, the number of microcracks also increases, and these move along the length of the fibers throughout the matrix, which will reduce the mechanical properties.

### 3.2. Effect of Short Fiber Length on Performance

#### 3.2.1. Mechanical Properties Test

In the process of filament formation, the grinding mixture and the screw extrusion process damage the fiber length. In this study, the SFs are directly added between the layers by laying, meaning that the SFs length is preserved in the test specimen to the greatest extent. The lengths of the SFs in the specimens were 0.1 mm, 0.5 mm, 1 mm and 2 mm, and the SFs content was 0.5 wt%. The tensile properties of the specimens are shown in Figure 11. When the length of the single-layer SFs is 0.5 mm, the average tensile strength of the specimen is 37.76 MPa, which is 23.20% higher than that of the pure resin specimen without adding SFs. A *t*-test was used to compare the tensile strength of the two groups (*p* = 0.003, *p* < 0.05 and *p* < 0.01, respectively). As the fiber length increases to 1 mm, the average tensile strength is 37.71 MPa, and the change is not obvious. When the fiber length increases up to 2 mm, the tensile strength rapidly decreases to 33.44 MPa, which is 11.32% lower than the extreme value. Figure 11b shows the changing trend of Young’s modulus with fiber length. When the fiber lengths are 0.5 mm and 1mm, the average value of Young’s modulus is 1.57 GPa. While the fiber length continues to increase to 2 mm, the modulus reduces to 1.20 GPa. Figure 12 shows the tensile stress-strain curves of two specimens with fiber lengths of 0.1 mm and 0.5 mm, respectively. As the strain increases, the specimen with the carbon fiber length of 0.5 mm first reaches the maximum tensile stress compared with the carbon fiber length of 0.1 mm. This shows that specimen with a carbon fiber length of 0.5 mm has a greater tensile strength and Young’s modulus, but poor toughness and ductility.

#### 3.2.2. Discussion

In the 3D printing process of traditional SFs reinforced composite materials, there are three steps of abrasion: fiber-screw, fiber-fiber and fiber-resin [4]. The three steps lead to large differences in the lengths of SFs between the filament and test specimen. Researchers have found that during experiments, the fiber length has an upper limit of about 0.4 mm [7]. In this study, the SFs are added by placement and length of the deposited fiber corresponds the length of the fiber in the forming part, the length of the fiber can be arbitrarily controlled. The testing results in Figure 11 and Figure 12 prove that SFs with a length of 0.5–1 mm can effectively improve the mechanical properties of the composite. Assuming that the interface between the fiber and the resin is well combined, from Equations (1)–(6), it can be seen that the load transfer length *L_f_* of the SFs does not increase with the increasing of the external load. When the force of the fiber reaches the maximum, the critical length *L_c_* is the minimum length corresponding to the maximum load-bearing efficiency of the fiber. When the fiber length is too short when mixing with the matrix, its performance cannot be fully exerted. According to formulas (7)–(11), when the fiber length is less than *L_c_*, the fiber will be pulled out when the fiber length is small, the stress is limited and transmitted between the fiber and the resin with restriction, and the enhancement effect is not obvious. The failure of the composite material depends on the failure of the matrix or interface. With the increasing of fiber length, the fibers will fracture instead of being pulled out when the specimen fail. When *L_f_* is larger than *L_c_*, the fiber plays the main bearing function. The performance of the material depends on the breaking of the fiber. When *L_f_* is much larger than *L_c_*, the mechanical properties of the composite material are similar to continuous fibers. In this study, when the fiber length is 0.5–1 mm, the fiber length is longer than the critical length, meaning that the mechanical properties of the composite material are optimal.

When the fiber length is 2 mm, the mechanical properties of the material are reduced. The main reason for this is that the fiber length of 2 mm is longer than the printing raster width of 1 mm. Excessively long fibers are prone to overlap and move, which leads to a decrease in the interfacial bonding quality between the fiber and resin. The problem of weak bonding is highlighted, weakening the load transfer capability between the SFs and the matrix. The failure mode of the composite material changes to the fiber being pulled out, and the mechanical properties are reduced. At this time, the SFs reinforced phase added by placement does not work. In summary, it is impracticable to simply increase the fiber length to improve the mechanical properties of the composite material. In further research, the interface bonding effect of the fiber and the resin will need to be improved to enhance the stress transmission ability at the interface.

### 3.3. Effect of Post-Treatment on Performance

#### 3.3.1. Mechanical Performance Test and SEM Observation

To improve the bonding effect between the SFs and the resin, the post-treatment of the formed specimens is performed. Table 3 shows the post-treatment temperature which is higher than the glass transition temperature. When the maximum post-treatment temperature is more than 150 °C, the rigidity of the specimen is insufficient, and deformation occurs under the action of gravity. A single-layer specimen with a fiber content of 0.5 wt% and fiber length of 0.5 mm was subjected to post-treatment at different temperatures. The test results of the mechanical properties are shown in Figure 13. The tensile strength and Young’s modulus changes nonlinearly with the increasing of the post-treatment temperature. When the post-treatment temperature is 140 °C, a micro-section is observed, as shown in Figure 14, which shows that the boundaries between the various printing layers are no longer clear, and the fusion phenomenon shown in Figure 14b occurs between the printed layers.

#### 3.3.2. XRD Testing

The crystallinity of PLA resin will change after post-treatment. The crystallinity of the specimen was tested by XRD; Figure 14 shows the testing results. The diffraction peak of the curve is sharp and the baseline is gentle, which is a typical crystallinity result for a polymer [25]. At diffraction angles 2*θ* of 16.7°, 19°, and 22.4°, clear diffraction peaks appear. The intensity of the diffraction peaks at different post treatment temperatures is very close to 2*θ*, indicating that the post-treatment does not change the crystal form of PLA, which is an *α* crystal form belonging with orthorhombic symmetry, which can be seen from the spectrum [26]. In Figure 15, Jade 5 software is used to perform fitting analysis on the map; then, the change of crystallinity with temperature is obtained. In the untreated specimen, a crystallinity of 17% was obtained. The crystallinity reaches a maximum value of 32% when post-treatment is performed at 100 °C. The crystallinity decreases when the post-treatment temperature is continuously increased.

#### 3.3.3. Discussion

The fiber-reinforced composite material made by additive manufacturing has poor interfacial bonding properties. The post-treatment was used to improve the bonding quality of the interlayer and the interface of fiber. Figure 16 shows the XRD results. On the one hand, the crystallinity of the specimen increases, the attraction between the molecular chains increases and the polymer chains are arranged closer together, which is consistent with previous post-treatment works for PLA composites [27]. On the other hand, the post-treatment temperature is higher than the glass transition temperature of the polymer, which causes the fusion phenomenon shown in Figure 14 between the printed layers. The procreant viscoelastic relaxation can release the residual thermal stress in the formed part to restore the interlayer tensile properties. Although the improvement of crystallinity and the fusion between the layers change the mechanical properties of the matrix and the bonding quality between the layers, the 5% improvement is not statistically significant. The improvement is not obvious, and further research on the post-treatment process is required to explore the mechanism of the performance optimization of the specimen.

### 3.4. Effect of Laser-Assisted Preheating on Performance

To further solve the problem of interfacial bonding between the deposited fiber and resin, this paper used laser-assisted preheating in additive manufacturing to form the specimen. When the fiber deposition is finished, the layer is scanned and preheated. Figure 17 shows the testing results of mechanical properties with different laser powers. The tensile strength of the specimen increases with the increase of the laser power. When the laser power is 10 W, the tensile strength of the specimen is 59.20 MPa, which is 56.78% higher than that of the unheated specimens. When the laser power is greater than 10 W, the resin surface is denatured by high temperature burning, so the maximum preheat power is selected to be 10 W. Figure 18 is a SEM micrograph of cross-sections with different laser powers. The fracture interface after preheating is rougher than the un-preheated specimen in Figure 11, in which a large number of fibers are coated by resin, and there are a large number of tensile fractures in the interface.

When the SFs are placed, the melt flowing of the resin is insufficient to fill the grooves and form a large number of defects form. The laser-assisted preheating method raises the interfacial temperature to the critical temperature increasing the interpenetrating diffusion of the resin. The impregnation effect of the fiber is also promoted. As the laser power increases, the interface preheating temperature and the local temperatures of the resin surface also increase, the resin fluidity improves, the impregnation is stronger and the interfacial bonding quality is improved. When the laser power is further increased, the decomposition point and vaporization point of the resin material are reached; then, defects are generated inside the specimen, decreasing the mechanical properties.

## 4. Conclusions

In this study, SFs were added to the interlayers of printed parts by placement, and the following conclusions were obtained by comparative analysis and experimental testing.

(a)When the fiber content was 0.5 wt% in a single layer, the tensile strength of the specimen reached 38 MPa, which is about 15% higher than that of the pure resin matrix. When the fiber content continuous to increase SFs the addition of SFs created many voids, resulting in a decrease in the mechanical properties.(b)When the fiber lengths were 0.5 mm and 1 mm, which are longer than the critical fiber length, the fiber played the main bearing role, and the composite had the best mechanical properties. As the fiber length increased to 2 mm, the fibers were liable to overlap and move, which resulted in a decrease of the bonding quality between interfaces.(c)Post-treatment was used on the specimens to improve the interfacial bonding quality. When the post-treatment temperature was higher than the glass transition temperature, fusion occurred between the printed layers, and the interface bonding quality was improved. However, the mechanical properties of the printed parts under different post-treatment conditions exhibited no statistically significant differences, and the range of mechanical properties was within 5%.(d)The method of laser-assisted preheating raised the interlayer temperature to the critical temperature, and then the interpenetrating diffusion of the resin increased, which promoted the impregnation between the fiber and resin. When the laser power was 10 W, the tensile strength of the laser-assisted preheated printed specimen was 59.20 MPa, which is an increase of 56.78% compared to the unheated specimen.

## Figures and Tables

**Figure 1 materials-13-02868-f001:**
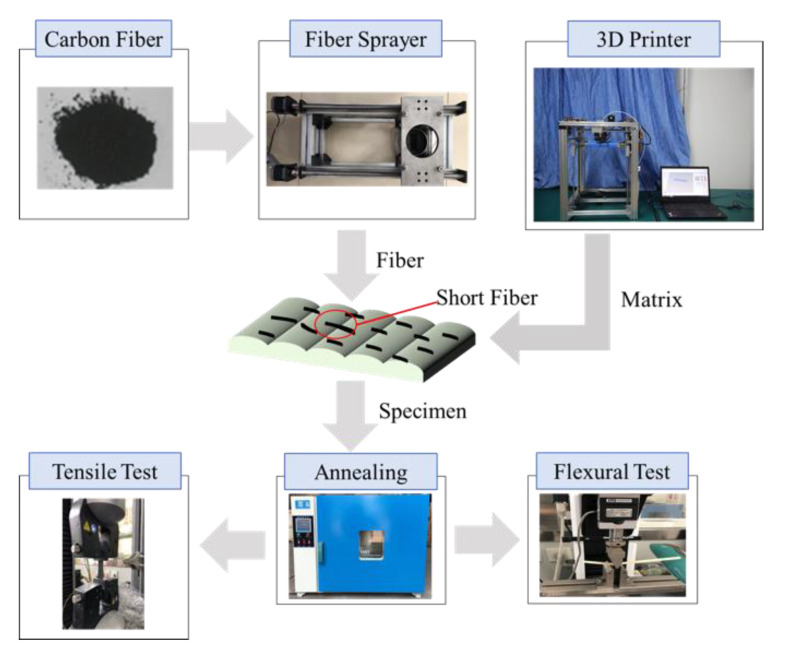
Short fibers (SFs) placement process in additive manufacturing (AM) and testing of specimens.

**Figure 2 materials-13-02868-f002:**
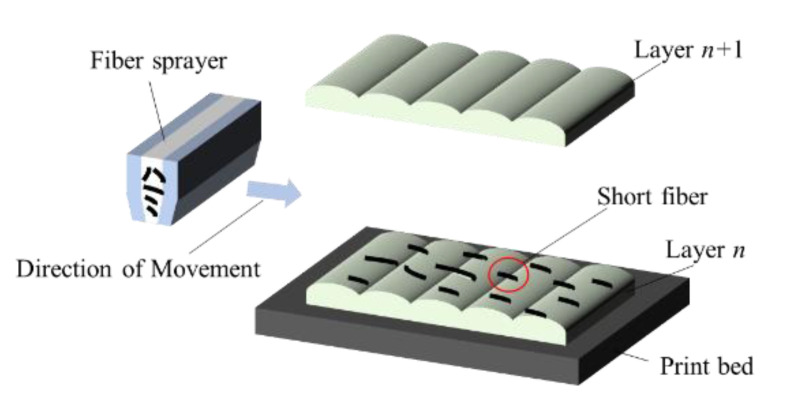
Schematic diagram of the inter-layer placement of SFs.

**Figure 3 materials-13-02868-f003:**
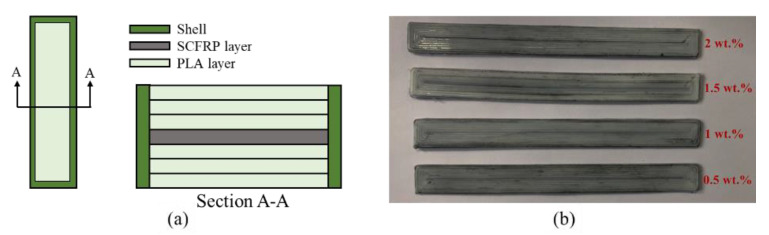
Test specimens of single-layer SFs: (**a**) schematic diagram; (**b**) physical picture. PLA: polylactic acid. SCFRP: short carbon fiber reinforced printing.

**Figure 4 materials-13-02868-f004:**
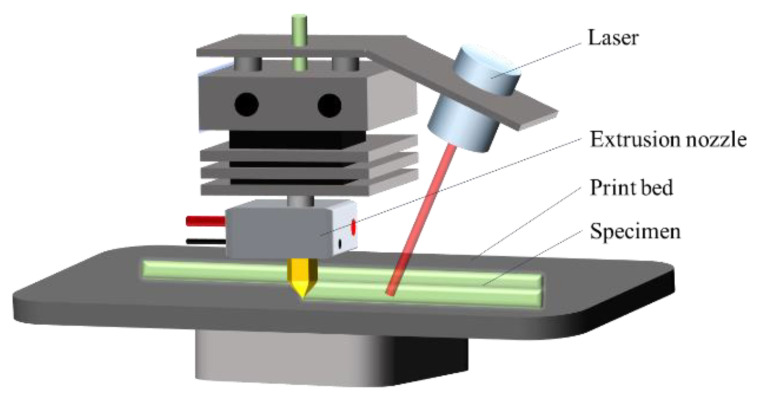
Schematic diagram of additive manufacturing with an integrated laser preheating function.

**Figure 5 materials-13-02868-f005:**
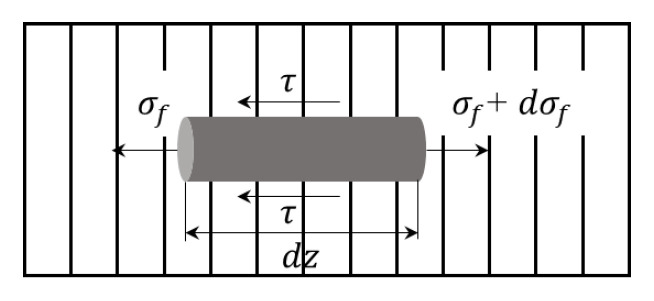
The stress state of the SFs in the resin matrix.

**Figure 6 materials-13-02868-f006:**
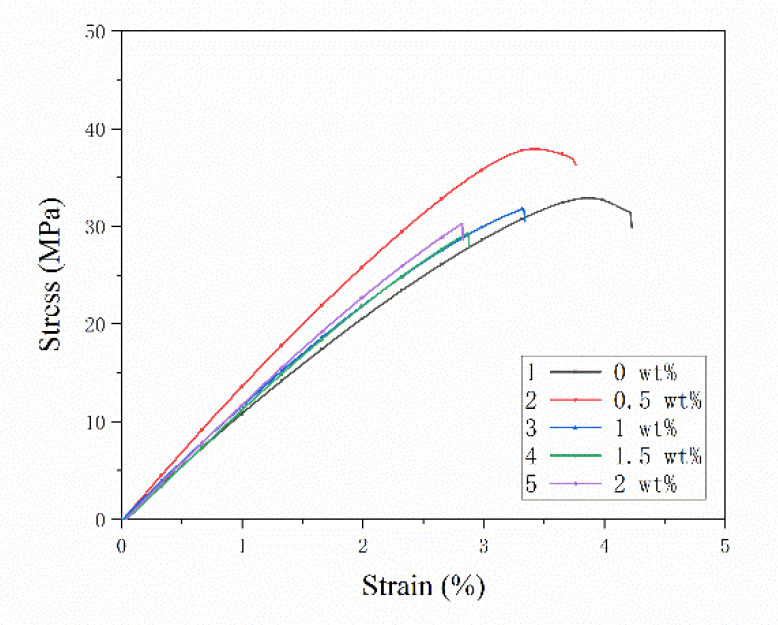
Typical tensile strain-stress curves for specimens with different carbon fiber contents (the carbon fiber length is 0.5 mm).

**Figure 7 materials-13-02868-f007:**
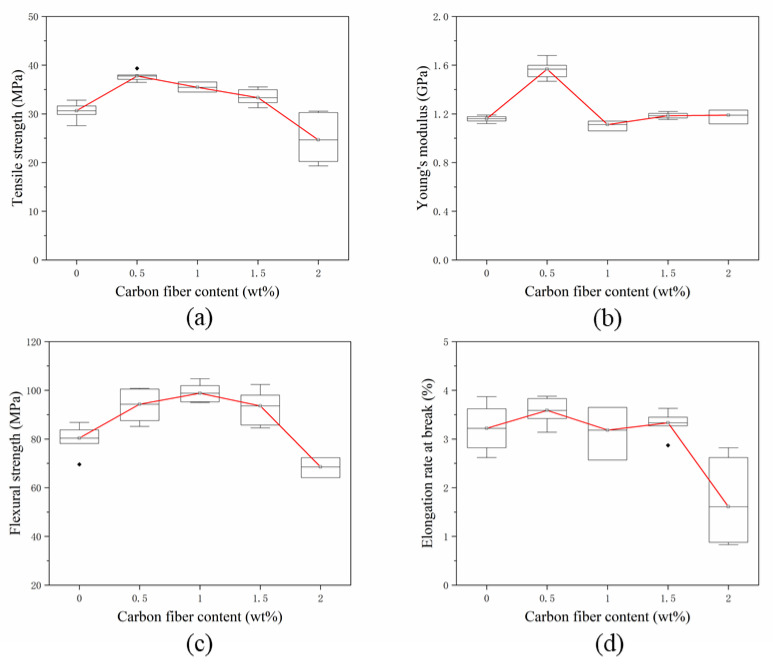
(**a**–**d**) Performance test of single-layer specimens with different carbon fiber contents.

**Figure 8 materials-13-02868-f008:**
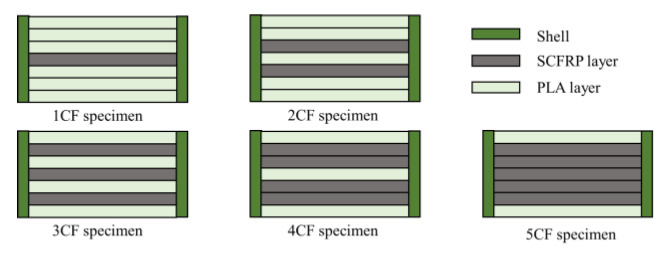
Different placement strategies for SFs.

**Figure 9 materials-13-02868-f009:**
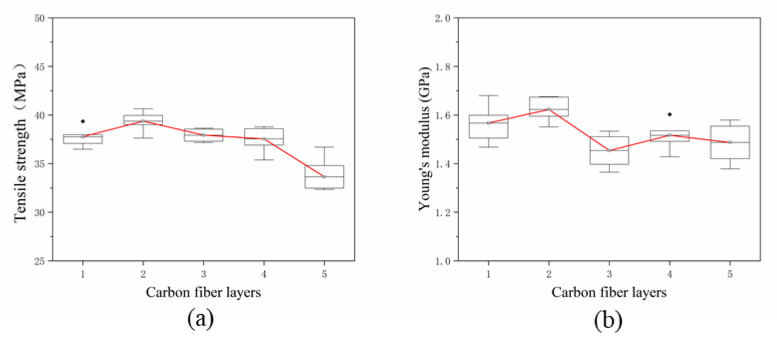
(**a**,**b**) Tensile properties of specimens with different SFs layers.

**Figure 10 materials-13-02868-f010:**
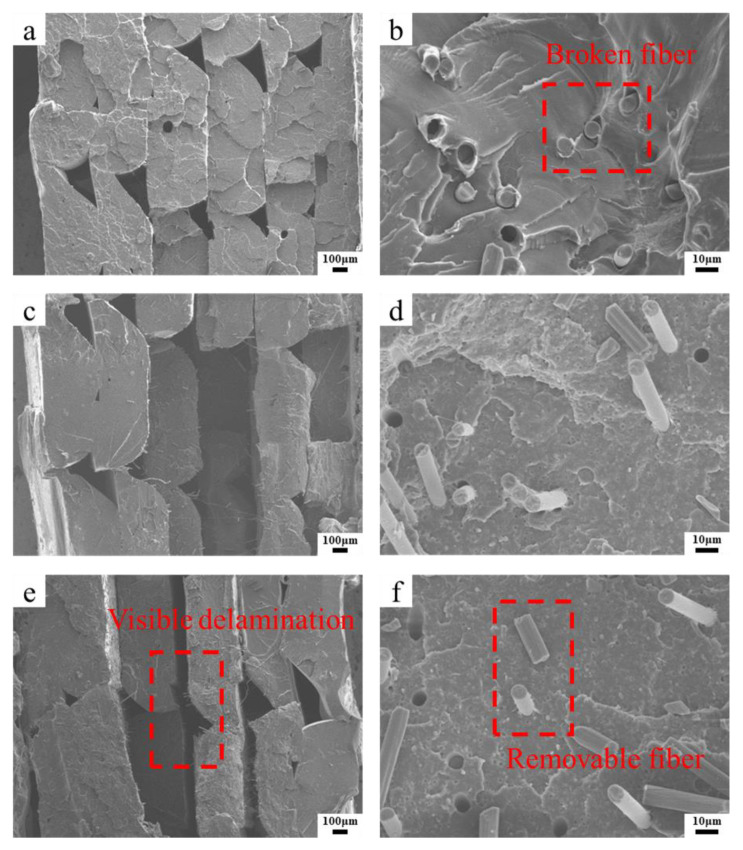
Microstructures of the fracture interface of cross section with different carbon fiber contents after tensile testing, respectively: (**a**), (**c**) and (**e**) overall cross section (50×), (**b**), (**d**) and (**f**) carbon fibers in the interface (750×), (**a**) and (**b**) fiber content 0.5 wt%, (**c**) and (**d**) fiber content 1.5 wt%,(**e**) and (**f**) fiber content 2 wt%.

**Figure 11 materials-13-02868-f011:**
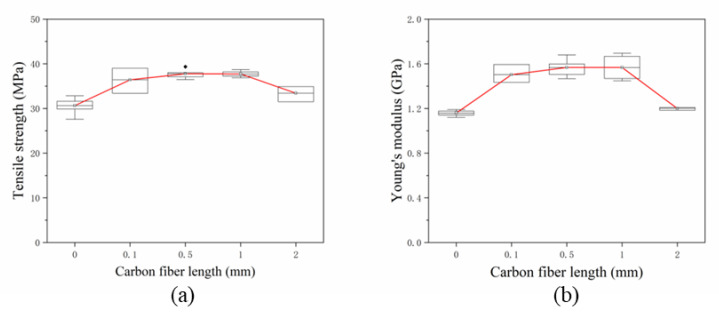
(**a**,**b**) Tensile properties of specimens with different fiber lengths.

**Figure 12 materials-13-02868-f012:**
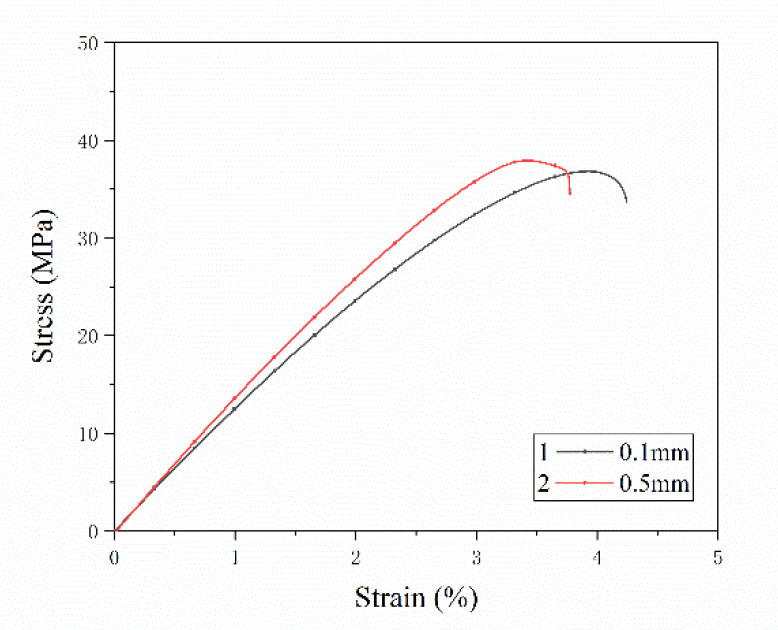
Stress-strain curve of specimens with different fiber lengths.

**Figure 13 materials-13-02868-f013:**
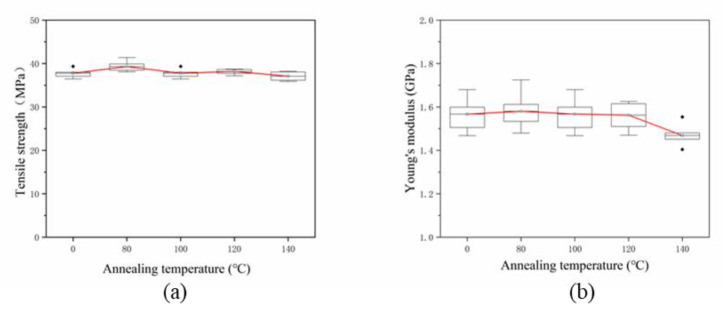
(**a**,**b**) Tensile properties of specimens with different post-treatment temperatures.

**Figure 14 materials-13-02868-f014:**
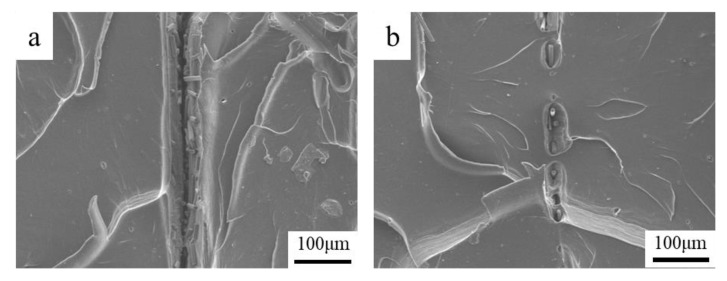
SEM micrographs of typical interlayer change before and after post-treatment (200×): (**a**) No post-treatment; (**b**) 140 °C post-treatment (interlayer fusion).

**Figure 15 materials-13-02868-f015:**
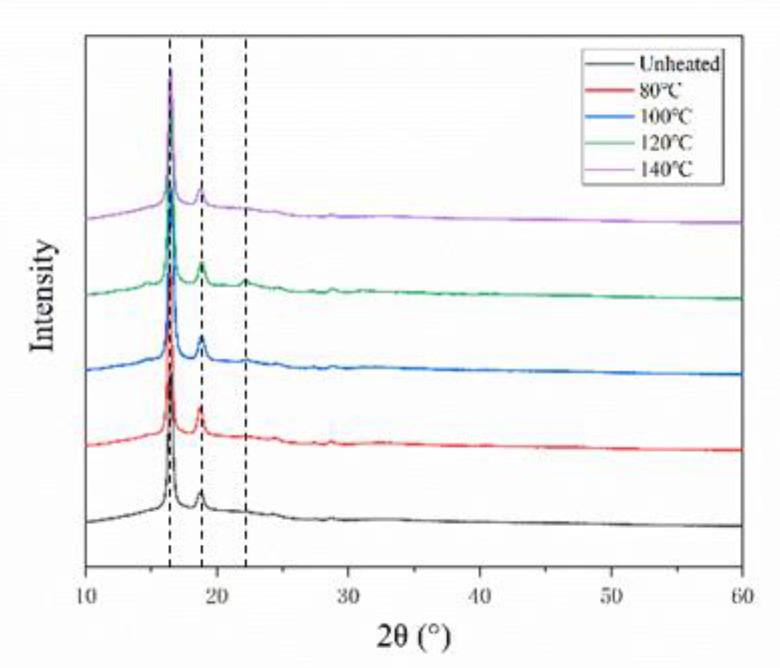
XRD results with different post-treatment temperatures.

**Figure 16 materials-13-02868-f016:**
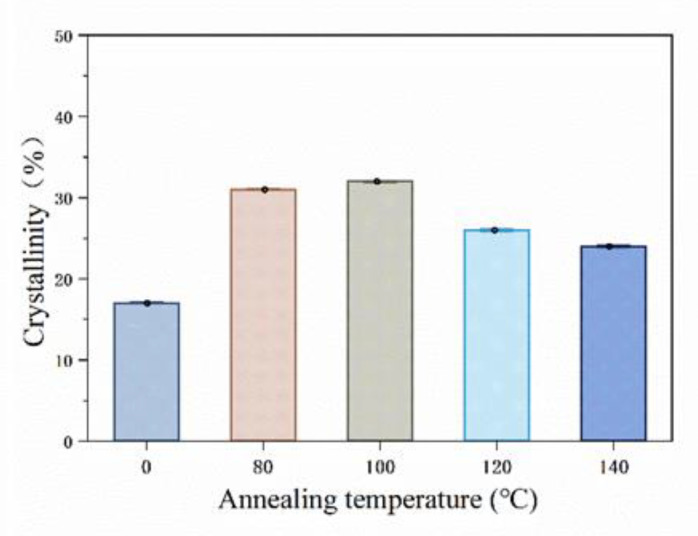
Crystallinity of specimens with different post-treatment temperatures.

**Figure 17 materials-13-02868-f017:**
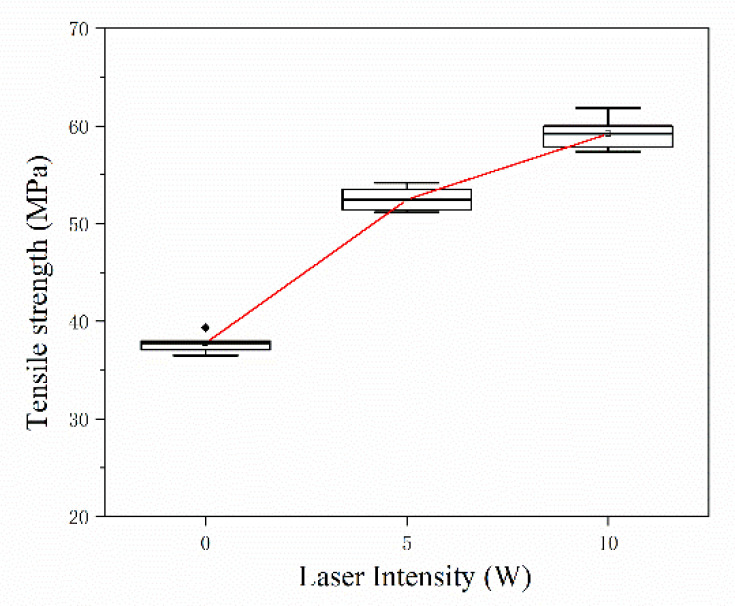
Tensile properties of specimens with different laser powers.

**Figure 18 materials-13-02868-f018:**
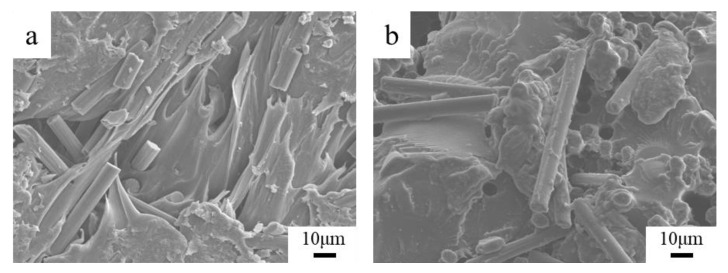
SEM micrographs of cross-section with different laser powers (750×): (**a**) 5 W; (**b**) 10 W.

**Table 1 materials-13-02868-t001:** Printing process parameters for specimens.

Parameters	Value
Hatch spacing—mm	1
Layer thickness—mm	0.5
Printing speed—mm/s	20
Diameter of nozzle—mm	1
Temperature of liquefier—°C	220
Infill density—%	100
Infill patterns	Concentric

**Table 2 materials-13-02868-t002:** Tensile properties test specimen, conforming to a Type geometry of GB/T 1447–2005–Standard.

Specimen Geometry Variable	Value	ISO GB/T 1447–2005–Type Geometry
Height (H)—mm	200	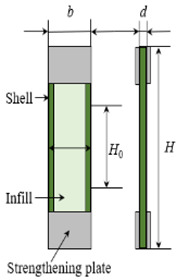
Width (b)—mm	18	
Thickness (d)—mm	3	
Standard distance (H_0_)—mm	50	
Number of shells (N_s_)	1	
Number of layers (N_l_)	6	

**Table 3 materials-13-02868-t003:** Annealing temperature for specimens. CF: carbon fiber.

Materials	Annealing Temperature (°C)	Number of Specimens
PLA + 0.5%CF	80	5
	100	5
	120	5
	140	5

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
