# Peer review of "Performance of Short Fiber Interlayered Reinforcement Thermoplastic Resin in Additive Manufacturing"

_materials, 2020, doi:10.3390/ma13122868_

Round 1
Reviewer 1 Report
The article "Performance of Short Fibers Interlayered Reinforcement Thermoplastic resin in Additive Manufacturing", authored by Congze Fan, Zhongde Shan, Guisheng Zou, Li Zhan and Dongdong Yan, describes an FDM based 3D printing technique adding the possibility to deposit short fibers in between the layers and presents an optimization of the process with respect to length and amount of fibers and post-processing of the printed parts. The manuscript would need considerable work-over (language and style) and I also have to raise severe doubts whether the experimental setup is suitable for an unbiased assessment of the performance of the process. Another severe drawback are the insufficiently described methods. In fact, in the article there is no technical information on how the dispension of short fibers actually works. Moreover, it remains unclear, what the "preheating" with the laser is meant to be because a "postheating" (after dispensing of short fibers) would rather make sense to me.
I was asked to speed up my review, so in the following I list the most important points, which led to my recommendation to reject the article:
1) A 3D printing technique is for one big part characterized by its possibility to print at high resolutions. FDM generally is one of the 3D printing techniques suffering from its comparable low resolution. In this article the nozzle size, which determines lateral resolution, was 1 mm! This means only very rough structures can me manufactured. The authors themselves explain that this large nozzles are necessary in order to make the process working (they state in the article that otherwise the layer on top of the fiber layer would not be deposited properly). This severe drawback is not at all discussed in the article!
2) The authors claim that mechanical tests were performed in accordance to ISO standards, however, the used specimens do not comply to the standards, e.g. the tensile testing bars are too short and too narrow. Moreover, dog bone specimens should be preferred and I really wonder why the authors did not choose to print appropriate specimens. Furthermore, I highly doubt that tensile tests are even suitable to assess the performance of the process (see next point).
3) Bending tests were only performed in the first series of tests and gave quite different results compared to the tensile tests. This is because bending tests are also affected by the interlayer phenomenons (which are predomenat in this kind of 3D printed parts) and I am convienced that delamination was observed here. For this reason bending test is the more unbiased method for characterization and should have been performed throughout the study!
4) In most of the cases the differences between the measured values are very low. The whole article lacks of statistical consideration of the obtained results. The p-values are necessary to discuss about the significance of the results.
Reviewer 2 Report
Review report
Manuscript ID: materials-823214
Title: Performance of Short Fibers Interlayered Reinforcement Thermoplastic resin in Additive Manufacturing
In this article, the authors have used short fibres as the reinforcements to enhance the performance of traditional thermoplastic resins. There are some points, which need to be clarified.
Please find the comments below:
- Page 1, section 1, Introduction part, line 32-33: “Among different types of resins, thermoplastic resins have unique characteristics, which are heat softening and cooling hardening [1-3] work.” – which different types of resins? This is not a good starting sentence for introduction.
- Page 2, Section 1, Introduction par, line 67-68: “Therefore, the low-cost, high-efficiency additive manufacturing of fiber reinforced composites are manufactured.”- Rephrase the sentence to give a clear expression.
- It is necessary to explain the abbreviation, ”SF” in the introduction. Further the ”Introduction” is not well organised. In the second paragraph, you have described the advantage of incorporation of SF into therplastic resin system, then in the third paragraph you have mentioned about the continuous fibres. The purpose of this work is not very clear. PLA-carbon fibre composites are widely reported so far. There is nothing new in using these raw materials.
I would suggest to reorganise the ”Introduction” part, so that there are smooth transitions between each paragraph. Also highlighting the differences between existing work and your work will be more appealing.
- Page 3, Section 2.2, line 98: Figure 1 caption: Where did you explain the term “FDM”?
What is the purpose of using annealing step in the work?
- In the experimental part, please mention the details on SEM: the magnification, accelerating voltage etc.
- Page 8, Section 3.1.2, line 251-252: “As shown in Figs. 10(a), (c) and (e), there are special triangle voids of 3D printing inside the specimen.”- how did these special triangle voids with ~same dimensions appear? Did this delamination happened during tensile measurement? Though the appearance of same pattern of voids is quite surprising. Is there any possibility of the slippage of the layers from one another during measurement? If so, which parameter contributed to this?
Overall the result discussion section is well explained. Experimental part should be explained in more detail and introduction part needs to be reorganized. So, the article could be considered for publication only after a major revision.
Reviewer 3 Report
Outline of article
This article investigated effects of direct application of short fibers on mechanical properties of printed parts. Also, studied influence of fiber distribution and interlayer bonding quality of the parts. Mechanical testing of printed parts revealed that an increase in properties for optimal direct reposition of fibers, and further pre-treatment and post-treatment of printed parts improved their overall mechanical performance.
Reviewer comments
Abstract:
Please do rewrite this section, content is novel but its not properly and selectively described. AM process is not specified.
e.g line 15, replace ‘deposited fiber length’ with length of fibers
line 16, distribution of fibers and quality of interlayer bonding were assessed using mechanical testing and microstructure examination.
Introduction:
Line 35, Do you mean that researcher have tried various methods to improve mechanical properties of printed parts.
Line 65, SF are selected as the reinforcement. Write main objective of the article …in first sentence and following that mention materials and methods employed to achieve that.
Line 66, poor mechanical properties in pure resin, do you mean that poor mechanical properties of parts printed with pure thermoplastic resin.
Also mention in this section, AM method you employed for you work
Entire introduction must go through English grammar checking, e.g. line 41, and process parameters in the additive manufacturing process- replaced with printing parameters
Line 42, by optimizing the nozzle temp, filling/printing speed and direction, and the ………., too long and not properly conveying its message
Line 50, how you relate that to work of ‘Mechanical behavior of 3D printed composite parts with short carbon fiber reinforcements’
Materials and methods:
Figure 1, not very clear. Clear process of direct SF applications must be illustrated, as this is a novel point in this article. Also, show clear picture of actual 3D printer used for building specimens
Also, preheating and post-treatment process must be described in details (parameters of the processes also), as they have greater influence on final mechanical properties of printed parts.
Line 120, not referenced properly to research of Marcus []?
Results and discussion
line 183, rewrite first paragraph of this section or delete, this is not giving any info of your research work,
section 3.1.1
Provide mechanical testing images; for tensile testing and flexural testing
discuss effect of CF reinforcements on mechanical properties and compare with results of specimens made of pure PLA resins and provide reasons for discrepancy in the results. How would compare these results with same observation made in the article ‘Mechanical behavior of 3D printed composite parts with short carbon fiber reinforcements’
Explicitly write paragraphs for single layer direction deposition of CF and multi-layer CF deposition, now line 229 is mixed with results of single layer CF specimens
SEM observation
Printing process inherent voids e.g. triangular voids, and poor interfacial bonding with increase in %CF causing delamination
Line 256, verify this again ‘breakage of CF during testing’ may not be true, as strength of the fibers is higher than the strength of PLA, the failure of specimen may be due only because of failure of PLA and bonding between PLA and CFs
Figure 10b, image of broken fiber may be one end of the actual fibers
discussion
Line 273, fiber breakage reason for failure of specimen may not be true as reasoned above
Effect of sCF length on performance
Figure 11b, Young’s modulus versus carbon fiber content, title of x- axis of the graph is it right?, is it not carbon fiber length?
Effect of laser assisted preheating on performance
Laser power is applied once the CF is deposited to improve interfacial bonding strength. 10 w laser power resulted highest tensile strength, more thana 10W laser damaging PLA resin.
Conclusion
Although summary of each section described here, but it is very lengthy. Please rewrite conclusion of each point succinctly.
Round 2
Reviewer 1 Report
The article "Performance of Short Fiber Interlayered Reinforcement Thermoplastic Resin in Additive Manufacturing" was resubmitted. The quality of manuscript has been improved considerably. I still recommend some proof reading as the text still contains some typing errors.
Reviewer 2 Report
Second Review Report
Manuscript ID: materials-823214
Title: Performance of Short Fiber Interlayered Reinforcement Thermoplastic Resin in Additive Manufacturing
In this manuscript, the authors have clarified all the points, I asked for, in my first review. The "Introduction" part has been modified as per the comments. Overall the article looks good and can be recommended for publication.
Reviewer 3 Report
the reviewer feels that the manuscript needs an extensive English grammar check, which was already suggested in the revision. Suggested revisions were not made in the manuscript.
